# ATOM3D:
# Tasks On Molecules in Three Dimensions

**Raphael J. L. Townshend**[*]
Computer Science
Stanford University

**Martin Vögele**[†]
Computer Science
Stanford University

**Patricia Suriana**[†]
Computer Science
Stanford University

**Alexander Derry**[†]
Biomedical Informatics
Stanford University

**Alexander S. Powers**
Chemistry
Stanford University

**Yianni Laloudakis**
Computer Science
Stanford University

**Sidhika Balachandar**
Computer Science
Stanford University

**Bowen Jing**
Computer Science
Stanford University

**Brandon Anderson**
Computer Science
University of Chicago

**Stephan Eismann**
Applied Physics
Stanford University

**Risi Kondor**
Computer Science, Statistics
University of Chicago

**Russ B. Altman**
Bioengineering, Genetics, Medicine
Stanford University

**Ron O. Dror**[*]
Computer Science
Stanford University

## Abstract

Computational methods that operate on three-dimensional (3D) molecular structure have the potential to solve important problems in biology and chemistry. Deep neural networks have gained significant attention, but their widespread adoption in the biomolecular domain has been limited by a lack of either systematic performance benchmarks or a unified toolkit for interacting with 3D molecular data. To address this, we present ATOM3D, a collection of both novel and existing benchmark datasets spanning several key classes of biomolecules. We implement several types of 3D molecular learning methods for each of these tasks and show that they consistently improve performance relative to methods based on one- and two-dimensional representations. The choice of architecture proves to be important for performance, with 3D convolutional networks excelling at tasks involving complex geometries, graph networks performing well on systems requiring detailed positional information, and the more recently developed equivariant networks showing significant promise. Our results indicate that many molecular problems stand to gain from 3D molecular learning, and that there is potential for substantial further improvement on many tasks. To lower the barrier to entry and facilitate further developments in the field, we also provide a comprehensive suite of tools for dataset processing, model training, and evaluation in our open-source `atom3d` Python package. All datasets are available for download from `www.atom3d.ai`.

---

[*]Address correspondence to: `raphael@cs.stanford.edu`, `rondror@cs.stanford.edu`
[†]Equal contribution.

35th Conference on Neural Information Processing Systems (NeurIPS 2021), Sydney, Australia.

Table 1: Representation choice for molecules. Adding in 3D information consistently improves performance. The depicted 1D representations are the amino acid sequences and SMILES strings [Weininger, 1988] for proteins and small molecules, respectively.

| Dimension | Representation | Examples | |
|---|---|---|---|
| | | Proteins | Small Molecules |
| 1D | linear sequence | KVKALPDA | CC(C)CC(C)NO |
| 2D | chemical bond graph | | |
| 3D | atomistic geometry | | |

# 1 Introduction

A molecule's three-dimensional (3D) shape is critical to understanding its physical mechanisms of action, and can be used to answer a number of questions relating to drug discovery, molecular design, and fundamental biology. While we can represent molecules using lower-dimensional representations such as linear sequences (1D) or chemical bond graphs (2D), considering the 3D positions of the component atoms—the atomistic geometry—allows for better modeling of 3D shape (Table 1). While previous benchmarking efforts, such as MoleculeNet [Wu et al., 2018] and TAPE [Rao et al., 2019], have examined diverse molecular tasks, they focus on these lower-dimensional representations. In this work, we demonstrate the benefit yielded by learning on 3D atomistic geometry and promote the development of 3D molecular learning by providing a collection of datasets leveraging this representation.

Furthermore, the atom is emerging as a "machine learning datatype" in its own right, deserving focused study much like the pixels that make up images in computer vision or the characters that make up text in natural language processing. All molecules, including proteins, DNA, RNA, and drugs, can be represented as atoms in 3D space. These atoms can only belong to a fixed class of element types (carbon, nitrogen, oxygen, etc.), and all molecules are governed by the same underlying laws of physics that impose rotational, translational, and permutational symmetries. These systems also contain higher-level patterns that are poorly characterized, creating a ripe opportunity for learning them from data: though certain basic components are well understood (e.g. amino acids, nucleotides, functional groups), many others can not easily be defined. These patterns are in turn composed in a hierarchy that itself is only partially elucidated.

While deep learning methods such as graph neural networks (GNNs) and convolutional neural networks (CNNs) seem well suited to atomistic geometry, to date there has been no systematic evaluation of such methods on molecular tasks. Additionally, despite the growing number of 3D structures available in databases such as the Protein Data Bank (PDB) [Berman et al., 2000], they require significant processing before they are useful for machine learning tasks. Inspired by the success of accessible databases such as ImageNet [Deng et al., 2009] and SQuAD [Rajpurkar et al., 2016] in sparking progress in their respective fields, we create and curate benchmark datasets for atomistic tasks, process them into a simple and standardized format, systematically benchmark 3D molecular learning methods, and present a set of best practices for other machine learning researchers interested in entering the field of 3D molecular learning (see Section C). We reveal a number of insights related to 3D molecular learning, including the relative strengths and weaknesses of different methods and the identification of several tasks that provide great opportunities for 3D molecular learning. These are all integrated into the atom3d Python package to lower the barrier to entry and facilitate reproducible research in 3D molecular learning for machine learning practitioners and structural biologists alike.

## 2 Related Work

While three dimensional molecular data have long been pursued as an attractive source of information in molecular learning and chemoinformatics [Swamidass et al., 2005, Azencott et al., 2007], their utility has become increasingly clear in the last couple years. Powered by increases in data availability and methodological advances, 3D molecular learning methods have demonstrated significant impact on tasks such as protein structure prediction [Senior et al., 2020, Jumper et al., 2021, Baek et al., 2021], equilibrium state sampling [Noé et al., 2019], and RNA structure prediction [Townshend et al., 2021]. At the same time, broader assessments of tasks involving molecular data have focused on either 1D or 2D representations [Wu et al., 2018, Rao et al., 2019]. Through ATOM3D, we aim to provide a first benchmark for learning on 3D molecular data. There are a few major classes of algorithms that exist for data in this form.

Graph neural networks (GNNs) have grown to be a major area of study, providing a natural way of learning from data with complex spatial structure. Many GNN implementations have been motivated by applications to atomic systems, including molecular fingerprinting [Duvenaud et al., 2015], property prediction [Schütt et al., 2017, Gilmer et al., 2017, Liu et al., 2019], protein interface prediction [Fout et al., 2017], and protein design [Ingraham et al., 2019]. Instead of encoding points in Euclidean space, GNNs encode their pairwise connectivity, capturing a structured representation of atomistic data. We note that some developed GNNs operate only on the chemical bond graph (i.e., 2D GNNs), with their edges representing covalent bonds, whereas others (including the ones we develop) operate on the 3D atomistic geometry (3D GNNs), with their edges representing distances between nearby pairs of atoms.

Three-dimensional CNNs (3DCNNs) have also become popular as a way to capture these complex 3D geometries. They have been applied to a number of biomolecular applications such as protein model quality assessment [Pagès et al., 2019, Derevyanko et al., 2018], protein sequence design [Anand et al., 2020], protein interface prediction [Townshend et al., 2019], and structure-based drug discovery [Wallach et al., 2015, Torng and Altman, 2017, Ragoza et al., 2017, Jiménez et al., 2018]. These 3DCNNs can encode translational and permutational symmetries, but incur significant computational expense and cannot capture rotational symmetries without data augmentation.

In an attempt to address many of the problems of representing atomistic geometries, equivariant neural networks (ENNs) have emerged as a new class of methods for learning from molecular systems. These networks are built such that geometric transformations of their inputs lead to well-defined transformations of their outputs. This setup leads to the neurons of the network learning rules that resemble physical interactions. Tensor field networks [Thomas et al., 2018] and Cormorant [Kondor, 2018, Anderson et al., 2019] have applied these principles to atomic systems and begun to demonstrate promise on larger systems such as proteins and RNA [Eismann et al., 2020, Weiler et al., 2018, Townshend et al., 2021].

## 3 Datasets for 3D Molecular Learning

We select 3D molecular learning tasks from structural biophysics and medicinal chemistry that span a variety of molecule types. Several of these datasets are novel, while others are extracted from existing sources (Table 2). We note that these datasets are intended for benchmarking machine learning representations, and do not always correspond to problem settings that would be seen in real-world scenarios. Below, we give a short description of each dataset's impact and source, as well as the metrics used to evaluate them and the splits. The splits were selected to minimize data leakage concerns and ensure generalizability and reproducibility. These datasets are all provided in a standardized format that requires no specialized libraries. Alongside these datasets, we present corresponding best practices (Appendix C) and further dataset-specific details (Appendix D). Taken together, we hope these efforts will lower the barrier to entry for machine learning researchers interested in developing methods for 3D molecular learning and encourage rapid progress in the field.

### 3.1 Small Molecule Properties (SMP)

**Impact** – Predicting physico-chemical properties of small molecules is a common task in medicinal chemistry and materials design. Quantum-chemical calculations can determine certain physico-chemical properties but are computationally expensive.

Table 2: Tasks included in the ATOM3D datasets, along with schematic representations of their inputs. P indicates protein, SM indicates small molecule, R indicates RNA. Lines indicate interaction and a small square within a protein indicates an individual amino acid. New datasets are in bold.

| Name (Task Code) | Schematic | Objective | Source |
|---|---|---|---|
| Small Molecule Properties (SMP) | SM | Properties | QM9 [Ruddigkeit et al., 2012] |
| Protein Interface Prediction (PIP) | P1 — P2 | Amino Acid Interaction | DIPS [Townshend et al., 2019] DB5 [Vreven et al., 2015] |
| **Residue Identity (RES)** | P | **Amino Acid Identity** | **New, created from PDB [Berman et al., 2000]** |
| **Mutation Stability Prediction (MSP)** | P1 — P2 vs. P1 — P2 | **Effect of Mutation** | **New, created from SKEMPI [Jankauskaitė et al., 2019]** |
| Ligand Binding Affinity (LBA) | P — SM | Binding Strength | PDBBind [Wang et al., 2004] |
| **Ligand Efficacy Prediction (LEP)** | P — SM vs. P — SM | **Ligand Efficacy** | **New, created from PDB [Berman et al., 2000]** |
| Protein Structure Ranking (PSR) | P | Ranking | CASP-QA [Kryshtafovych et al., 2019] |
| RNA Structure Ranking (RSR) | R | Ranking | FARFAR2-Puzzles [Watkins et al., 2020] |

**Source** – The QM9 dataset [Ruddigkeit et al., 2012, Ramakrishnan et al., 2014b] contains the results of quantum-chemical calculations for 134,000 stable small organic molecules, each made up C, O, N, F, and H and including no more than nine non-hydrogen atoms. For each molecule, the dataset contains the calculated geometry of the ground-state conformation as well as calculated energetic, electronic, and thermodynamic properties.
**Targets** – We predict the molecular properties from the ground-state structure.
**Split** – We split molecules randomly.

## 3.2 Protein Interface Prediction (PIP)

**Impact** – Proteins interact with each other in many scenarios—for example, antibody proteins recognize diseases by binding to antigens. A critical problem in understanding these interactions is to identify which amino acids of two given proteins will interact upon binding.
**Source** – For training, we use the Database of Interacting Protein Structures (DIPS), a comprehensive dataset of protein complexes mined from the PDB [Townshend et al., 2019]. We predict on the Docking Benchmark 5 [Vreven et al., 2015], a smaller gold standard dataset.
**Targets** – We predict whether two amino acids will contact when their respective proteins bind.
**Split** – We split protein complexes such that no protein in the training dataset has more than 30% sequence identity with any protein in the DIPS test set or the DB5 dataset.

### 3.3 Residue Identity (RES)

**Impact** – Understanding the structural role of individual amino acids is important for engineering new proteins. We can understand this role by predicting the propensity for different amino acids at a given protein site based on the surrounding structural environment [Torng and Altman, 2017].

**Source** – We generate a novel dataset consisting of local atomic environments centered around individual residues extracted from non-redundant structures in the PDB.

**Targets** – We formulate this as a classification task where we predict the identity of the amino acid in the center of the environment based on all other atoms.

**Split** – We split environments by protein topology class according to the CATH 4.2 [Dawson et al., 2017], such that all environments from proteins in the same class are in the same split dataset.

### 3.4 Mutation Stability Prediction (MSP)

**Impact** – Identifying mutations that stabilize a protein's interactions is important to the design of new proteins. Experimental techniques for probing such mutations are labor-intensive [Antikainen and Martin, 2005, Lefèvre et al., 1997], motivating the development of efficient computational methods.

**Source** – We derive a novel dataset by collecting single-point mutations from the SKEMPI database [Jankauskaitė et al., 2019] and model each mutation into the structure to produce a mutated structure.

**Targets** – We formulate this as a binary classification task where we predict whether the stability of the complex increases as a result of the mutation.

**Split** – We split protein complexes such that no protein in the test dataset has more than 30% sequence identity with any protein in the training dataset.

### 3.5 Ligand Binding Affinity (LBA)

**Impact** – Predicting the strength (affinity) of a candidate drug molecule's interaction with a target protein is a challenging but crucial task for drug discovery applications.

**Source** – We use the PDBBind database [Wang et al., 2004, Liu et al., 2015], a curated database containing protein-ligand complexes from the PDB and their corresponding binding strengths (affinities).

**Targets** – We predict $pK = -\log_{10}(K)$, where $K$ is the binding affinity in Molar units.

**Split** – We split protein-ligand complexes such that no protein in the test dataset has more than 30% sequence identity with any protein in the training dataset.

### 3.6 Ligand Efficacy Prediction (LEP)

**Impact** – Many proteins switch on or off their function by changing shape. Predicting which shape a drug will favor is thus an important task in drug design.

**Source** – We develop a novel dataset by curating proteins from several families with both "active" and "inactive" state structures, and model in 527 small molecules with known activating or inactivating function using the program Glide [Friesner et al., 2004].

**Targets** – We formulate this as a binary classification task where we predict whether a molecule bound to the structures will be an activator of the protein's function or not.

**Split** – We split complex pairs by protein target.

### 3.7 Protein Structure Ranking (PSR)

**Impact** – Proteins are one of the primary workhorses of the cell, and knowing their structure is often critical to understanding (and engineering) their function.

**Source** – We use the structural models submitted to the Critical Assessment of Structure Prediction (CASP) [Kryshtafovych et al., 2019], a blind protein structure prediction competition, over the last 18 years.

**Targets** – We formulate this as a regression task, where we predict the global distance test (GDT_TS) of each structural model from the experimentally determined structure.

**Split** – We split structures temporally by competition year.

### 3.8 RNA Structure Ranking (RSR)

**Impact** – Similar to proteins, RNA plays major functional roles (e.g., gene regulation) and can adopt well-defined 3D shapes. Yet the problem is data-poor, with only a few hundred known structures.
**Source** – We use the FARFAR2-Puzzles dataset, which consists of structural models generated by FARFAR2 [Watkins et al., 2020] for 20 RNAs from RNA Puzzles, a blind structure prediction competition for RNA [Cruz et al., 2012].
**Targets** – We predict the root-mean-squared deviation (RMSD) of each structural model from the experimentally determined structure.
**Split** – We split structures temporally by competition year.

## 4  Benchmarking Setup

To assess the benefits of 3D molecular learning, we use a combination of existing and novel 3D molecular learning methods, and implement a number of robust baselines. Our 3D molecular learning methods belong to one of each of the major classes of deep learning algorithms that have been applied to atomistic systems: graph networks, three-dimensional convolutional networks, and equivariant networks. Here we describe the main principles of the core networks used in these models. See Appendix E for task-specific details and hyperparameters.

For GNNs, we represent molecular systems as graphs in which each node is an atom. Edges are defined between all atoms separated by less than $4.5$ Å, and weighted by the distance between the atoms using an edge weight defined by $w_{i,j} = \frac{1}{d_{i,j}+\epsilon}$, where $\epsilon = 10^{-5}$ is a small factor added for numerical stability. Node features are one-hot-encoded by atom type. Our core model uses five layers of graph convolutions as defined by Kipf and Welling [2016], each followed by batch normalization and ReLU activation, and finally two fully-connected layers with dropout. For tasks requiring a single output for the entire molecular system, we use global pooling to aggregate over nodes. For tasks requiring predictions for single atoms or amino acids, we extract the relevant node embeddings from each graph after all convolutional layers (see Appendix E).

For 3DCNNs, we represent our data as a cube of fixed size (different per task due to the different molecular sizes) in 3D space that is discretized into voxels with resolution of $1$ Å to form a grid (for PSR and RSR, we decrease the grid resolution to $1.3$ Å in order to fit in the GPU memory). Each voxel is associated with a one-hot-encoded vector that denotes the presence or absence of each atom type. Our core model consists of four 3D-convolutional layers, each followed by ReLU activation and max-pooling (for every other convolution layer). Two fully connected layers are applied after the convolutional layers to produce the final prediction.

For ENNs, we use SE(3)-equivariant networks that represent each atom of a structure by its coordinates in 3D space and by a one-hot encoding of its atom type. No rotational augmentation is needed due to the rotational symmetry of the network. The core of all architectures in this work is Cormorant, a network of four layers of covariant neurons that use the Clebsch–Gordan transform as nonlinearity, as described and implemented by Anderson et al. [2019].

## 5  Benchmarking Results

To assess the utility of 3D molecular learning, we evaluate our methods on the ATOM3D datasets and compare performance to state-of-the-art methods using 1D or 2D representations (for a comparison to the overall state-of-the-art, see Table 7). We note that in many cases, 3D molecular learning methods have not been applied to the proposed tasks, and that several of the tasks are novel. In the following sections, we describe the results of our benchmarking and some key insights that can be derived from them. We also aggregate these results along with additional metrics and standard deviations over three replicates in Table 8. For each metric, we bold the best-performing method as well as those within one standard deviation of the best-performing method.

### 5.1  3D representations consistently improve performance

Our evaluation of 3D methods on the tasks in ATOM3D reveals that incorporating atomistic geometry leads to consistently superior performance compared to 1D and 2D methods. For small molecules,

Table 3: Small molecule results. Metric is mean absolute error (MAE).

| Task | Target | 3D | | | Non-3D | |
| --- | --- | --- | --- | --- | --- | --- |
| | | 3DCNN | GNN | ENN | [Tsubaki et al., 2019] | [Liu et al., 2019] |
| SMP | $\mu$ [D] | 0.754 | 0.501 | **0.052** | 0.496 | 0.520 |
| | $\varepsilon_{\mathrm{gap}}$ [eV] | 0.580 | 0.137 | **0.095** | 0.154 | 0.184 |
| | $U_0^{\mathrm{at}}$ [eV] | 3.862 | 1.424 | **0.025** | 0.182 | 0.218 |

Table 4: Biopolymer results. AUROC is the area under the receiver operating characteristic curve. Asterisks (*) indicate that the exact training data differed (though splitting criteria were the same).

| Task | Metric | 3D | | | Non-3D |
| --- | --- | --- | --- | --- | --- |
| | | 3DCNN | GNN | ENN | [Sanchez-Garcia et al., 2018] |
| PIP | AUROC | **0.844** | *0.669 | — | 0.841 |
| | | | | | [Rao et al., 2019] |
| RES | accuracy | **0.451** | 0.082 | *0.072 | *0.30 |
| MSP | AUROC | 0.574 | **0.609** | 0.574 | 0.554 |

state-of-the-art methods do not use 1D representations, so we focus instead on comparing to representations at the 2D level, i.e. the chemical bond graph. This is the approach taken by the 2D GNN introduced by [Tsubaki et al., 2019] and the N-gram graph method by [Liu et al., 2019], which both obtain similar results (Table 3) on the small-molecule-only dataset SMP. When we add 3D coordinate information as in our ENN implementation, performance improves across all targets in SMP.

For tasks involving biopolymers (proteins and RNA), state-of-the-art methods do not use 2D representations, primarily because most of the chemical bond graph can be re-derived from the 1D representation, i.e. the linear sequence that makes up the biopolymer. We thus compare to representations at the 1D level (Table 4). For MSP and RES, both new datasets, we evaluate against the TAPE model [Rao et al., 2019], a transformer architecture that operates on protein sequence and is state-of-the-art amongst 1D methods for many tasks. For PIP, we compare to the sequence-only version of BIPSPI [Sanchez-Garcia et al., 2018], a state-of-the-art boosted decision tree method for protein interaction prediction. We find that 3D methods outperform these 1D methods on all biopolymer-only datasets (PIP, RES, MSP).

For tasks involving both biopolymers and small molecules, we compare to DeepDTA [Öztürk et al., 2018]. This network uses a 1D representation via a 1DCNN for both the biopolymer and small molecules. For LBA, we additionally compare to DeepAffinity [Karimi et al., 2019] which uses pairs of ligand SMILES strings and structurally annotated protein sequences. Using a 3D representation for both the ligand and protein leads to improved or comparable performance for the joint protein–small molecule datasets (LBA and LEP, see Table 5).

The biopolymer structure ranking tasks (PSR and RSR) are inherently 3D in nature, as they involve evaluating the correctness of different 3D shapes taken on by the same biopolymer. Thus, critically,

Table 5: Joint small molecule/biopolymer results. $R_S$ is Spearman correlation, $R_P$ is Pearson correlation, AUROC is area under the receiver operating characteristic curve, and RMSE is root-mean-squared error.

| Task | Metric | 3D | | | Non-3D | |
| --- | --- | --- | --- | --- | --- | --- |
| | | 3DCNN | GNN | ENN | [Öztürk et al., 2018] | [Karimi et al., 2019] |
| LBA | RMSE | **1.416** | 1.601 | 1.568 | 1.565 | 1.893 |
| | glob. $R_P$ | 0.550 | 0.545 | 0.389 | **0.573** | 0.415 |
| | glob. $R_S$ | 0.553 | 0.533 | 0.408 | **0.574** | 0.426 |
| LEP | AUROC | 0.589 | **0.681** | 0.663 | **0.696** | — |

Table 6: Structure ranking results. $R_S$ is Spearman correlation. Mean measures the correlation for structures corresponding to the same biopolymer, whereas global measures the correlation across all biopolymers.

| Task | Metric | 3D | | |
|------|--------|-------|-------|------|
| | | 3DCNN | GNN | SotA |
| PSR | mean $R_S$ | **0.431** | 0.411 | **0.432** [Pagès et al., 2019] |
| | glob. $R_S$ | **0.789** | 0.750 | **0.796** [Pagès et al., 2019] |
| RSR | mean $R_S$ | **0.264** | **0.234** | 0.173 [Watkins et al., 2020] |
| | glob. $R_S$ | 0.372 | **0.512** | 0.304 [Watkins et al., 2020] |

a 1D or 2D representation would not be able to differentiate between these different shapes since the linear sequence and chemical bond graph would remain the same. We therefore compare to state-of-the-art 3D methods as shown in Table 6, finding competitive or better results.

More generally, we find that learning methods that leverage the 3D geometry of molecules hold state-of-the-art on the majority of tasks on our benchmark (Table 7). These results demonstrate the potential of 3D molecular learning to address a wide range of problems involving molecular structure, and we anticipate that continued development of such models will aid progress in biological and chemical research.

## 5.2 Different tasks benefit from different architectures

While 3D molecular learning methods outperform their non-3D counterparts and provide a systematic way of representing molecular data, our results also provide evidence that architecture selection plays an important role in performance.

For tasks focused on biopolymers and with large amounts of training data (PIP and RES, Figure 1) we observe that 3DCNNs generally outperform standard GNNs. We hypothesize this is due to the ability of 3DCNNs to learn many-body patterns within a single filter, as opposed to GNNs that operate on one-body (node) and two-body (edge) features. Such many-body patterns are especially present in biopolymers, which generally adopt complex 3D geometries. This *many-body representation* hypothesis implies that 3DCNNs have specific advantages in terms of representational power.

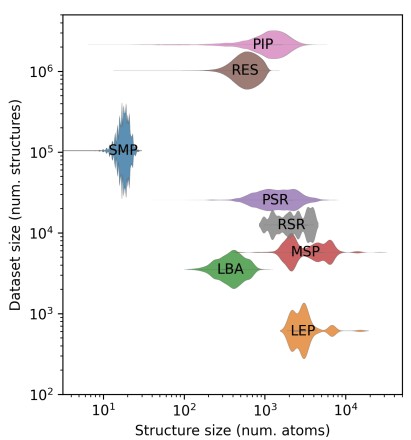

Figure 1: Datasets plotted by their size, as well as the number of atoms in each one of their molecules.

However, as the size of the datasets decrease (Table 9), we see more even performance when comparing 3DCNNs and GNNs. In particular, performance is quite similar on the intermediate-sized PSR and RSR datasets, and GNNs nearly fully supplant 3DCNNs on the small-sized MSP and LEP datasets. On these datasets, ENNs also equal or exceed 3DCNN performance. This is in line with the many-body representation hypothesis, as the increased representational power of 3DCNNs becomes less important in these data-poor regimes.

A notable exception to this trend is the large SMP dataset, where we see improved performance from GNNs and ENNs. We note, however, that this is the sole dataset involving only small molecules. These molecules generally do not contain as complex of 3D geometries as biopolymers, and therefore do not contain large numbers of many-body patterns. In addition, many of the prediction targets depend instead on the exact positions of each atom relative to its neighbors. While particle-based methods such as GNNs and ENNs can precisely record these distances, volumetric 3DCNNs must instead approximate these positions. While increasing spatial resolution increases precision, it also leads to cubic scaling of complexity that prevents the same level of precision.

Finally, equivariant networks show significant promise, despite being a recent innovation. One motivation for their use is that they fill a "happy medium" where they both represent atom positions precisely and capture the many-body patterns present in complex geometries. On SMP, the only dataset on which we tested ENNs without any limitations, we observed state-of-the-art performance. For other tasks, the performance of the ENN implementation we used limited us to training on a fraction of the data ($< 1\%$ for RES) or on a portion of the entire atomic structure (LBA, LEP, MSP), or did not permit us to apply it at all (PIP, PSR, RSR). Faster implementations are now available to allow scaling of ENNs to larger systems [Geiger et al., 2020, Kondor and Thiede, 2021].

# 6    Conclusion

In this work we present a vision of the atom as a new "machine learning datatype" deserving focused study, as 3D molecular learning has the potential to address many unsolved problems in biology and chemistry. In particular, systems of atoms are well-suited to machine learning as they contain several underlying symmetries as well as poorly understood higher-level patterns. With ATOM3D, we take a first step towards this vision by providing a comprehensive suite of benchmark datasets and computational tools for building machine learning models for 3D molecular data.

We provide several benchmark datasets and compare the performance of different types of 3D molecular learning models across these tasks. We demonstrate that, for nearly all tasks that can be formulated in lower dimensions, 3D molecular learning yields gains in performance over 1D and 2D methods. We also show that selection of an appropriate architecture is critical for optimal performance on a given task; depending on the structure of the underlying data, a 3DCNN, GNN, or ENN may be most appropriate, especially in light of our many-body representation hypothesis. Equivariant networks in particular are continuing to improve in efficiency and stability, and we expect these to prove effective due to their ability to concisely model physical laws.

While this work demonstrates the potential of 3D structures and provides an initial set of benchmarks, there are some limitations to consider when using these resources. First, the datasets and tasks represented in ATOM3D are inherently biased towards biomolecules with solved structures. Certain classes of molecules (e.g. intrinsically disordered or transmembrane proteins) may therefore be underrepresented or absent, and the performance on these benchmarks will not necessarily generalize to such structures. Second, the benchmark models we report here are designed to be competitive but simple baselines. A bespoke architecture designed specifically for a certain task or molecule class and with comprehensive hyperparameter tuning is expected to outperform many of these baselines, and we encourage the exploration of novel and innovative approaches even within model classes that appear to underperform in these benchmarks (e.g. GNNs for the PIP task).

Third, several of the benchmark tasks are formulated differently from those that biologists, chemists, and drug designers typically wish to solve. For example, when predicting the binding affinity of a ligand, one would rarely have access to a 3D structure of the ligand bound to the target (as in the LBA benchmark), because determining this structure experimentally would typically be far more expensive and time-consuming than measuring the ligand binding affinity. Likewise, when using machine learning methods to predict small-molecule properties more efficiently than through quantum chemical calculations (as in the SMP benchmark), one would not typically have access to the ground-state structure, because determining that structure requires equally expensive quantum chemical calculations. Although formulated in a somewhat artificial manner for convenience, such benchmarks have proven useful in evaluating general machine learning representations and methods.

Finally, in addition to the datasets described here, there are many other open areas in biomedical research and molecular science that are ripe for 3D molecular learning, especially as structural data becomes readily available. Such tasks include virtual screening and pose prediction of small molecule drug candidates, as well as the incorporation of conformational ensembles instead of static structures in order to represent more faithfully the entire set of structures a molecule could adopt. Building on our easily extensible framework, we anticipate the addition of new datasets and tasks from across the research community.

Through this work, we hope to lower the entry barrier for machine learning practitioners, encourage the development of algorithms focused on 3D atomistic data, and promote an emerging paradigm within the fields of structural biology and medicinal chemistry.

# 7 Acknowledgments

We thank Truong-Son Hy, Maria Karelina, David Liu, Lígia Melo, Joseph Paggi, and Erik Thiede for discussions and advice. We also thank Aditi Krishnapriyan and Nicolas Swenson for pointing out an error in earlier GNN performance numbers. This work was supported by the U.S. Department of Energy (DOE), Office of Science, Graduate Student Research (SCGSR) program (RJLT); EMBO Long-Term Fellowship ALTF 235-2019 (MV); an NSF Graduate Research Fellowship (PS); National Library of Medicine training grant LM012409 (AD); a Stanford Bio-X Bowes Fellowship (SE); NIH grants GM102365 and HG010615 (RBA); the Chan Zuckerberg Biohub (RBA); the DOE, Office of Science, Scientific Discovery through Advanced Computing (SciDAC) program (ROD); Intel (ROD); and DARPA Agreement No. HR0011-18-9-0038 (RK). Most of the computing for this project was performed on the Sherlock cluster. We thank Stanford University and the Stanford Research Computing Facility for providing computational resources and support that contributed to these research results.

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
