# OpenReview forum: "ATOM3D: Tasks on Molecules in Three Dimensions"
_NeurIPS.cc/2021/Track/Datasets_and_Benchmarks/Round1 — NeurIPS 2021 Datasets and Benchmarks Track (Round 1)_

### Official Review · Reviewer_RgsX · 2021-06-28
**A useful resource that is accessible by machine learning practitioners**

**Rating:** 8
**Confidence:** 3
**Correctness:** Yes.
**Clarity:** Overall clear but could be improved (…

**Strengths:**

+ The presented ATOM3D benchmark is a useful resource to the community. Notably, it makes 3D molecular learning extremely accessible by integrating everything into the atom3d Python packages. It is valuable in accelerating research into this field.
+ There are a number of insights about the benefits of using 3DCNN and GNN over conventional 2D and 1D methods on multiple tasks. These findings and insights are useful to know.
+ It identifies opportunities in several under-explored molecular tasks.

**Weaknesses:**

- I think some of the phrasing in the paper is obscure and needs improvement from a machine learning practitioner's standpoint. For example, it is not very clear if line 173 is describing the metric or the task. "We predict the root-mean-squared deviation (RMSD) from the ground truth structure".  RMSD seems to be a metric, but how do you predict a metric? I think what it meant was to "evaluate the predictions on the RMSD metric".

- It is not clear what we can learn from some of the results presented in table 6. For example, it is not clear what is the difference between 3DCNN and the SotA of [Alford,2017], although both methods are 3D. Simply showing one is better does not tells us much about why one outperforms another.

- I think it helps to give more stats about each dataset presented in the benchmarks, e.g. number of samples etc.

**Additional Feedback:**

==============================
Post-rebuttal:
I appreciate the authors for addressing my concerns. I am raising my rating.

**Documentation:**

Yes

**Relation To Prior Work:**

Yes

**Summary And Contributions:**

This paper presents ATOM3D, a collection of both novel and existing benchmark datasets spanning several key classes of biomolecules. It experiments extensively with 3D molecular learning methods on various tasks presented in ATOM3D, and concludes a set of best practices for the field of 3D molecular learning, and reveals a number of insights related to 3D molecular learning. More notably, it makes 3D molecular learning extremely accessible by integrating everything into the atom3d Python packages.

---

> ### Author Response · Authors · 2021-07-14
> **Reviewer Response**
>
> We thank the reviewer for their comments and suggestions as to how to improve the paper!  We respond to these below, as well as with edits to the paper.
>
> > I think some of the phrasing in the paper is obscure and needs improvement from a machine learning practitioner's standpoint. For example, it is not very clear if line 173 is describing the metric or the task. "We predict the root-mean-squared deviation (RMSD) from the ground truth structure". RMSD seems to be a metric, but how do you predict a metric? I think what it meant was to "evaluate the predictions on the RMSD metric".
>
> Thank you for the comment. We agree our use of the term “metric” and “RMSD” can lead to confusion.  In Section 3, we are presenting the target value we are trying to predict, not the evaluation metric we are using. For PSR and RSR, RMSDs are indeed the targets of prediction. RMSD in this case is a quantitative measure of the similarity between two superimposed atomic coordinates; it measures the deviation of a structure from the ground truth (crystallographic) structure. To avoid confusion, we have renamed “metrics” in Section 3 to “targets”, and renamed “RMSD” to “RMSE” where it is used as an evaluation metric.
>
> >It is not clear what we can learn from some of the results presented in table 6. For example, it is not clear what is the difference between 3DCNN and the SotA of [Alford,2017], although both methods are 3D. Simply showing one is better does not tell us much about why one outperforms another.
>
> This is a fair point—there are a lot of open questions as to what features are driving these methods' relative performances.  We note that we briefly discuss the 3D SotAs from Table 6 in Section 5.2. In particular, the 3D SoTA for PSR [Pagès et al., 2019] is a 3DCNN method that is comparable in performance to our baseline 3DCNN model, and the 3D SotA for RSR [Alford et al., 2017] is based on Rosetta all-atom energy function, which uses handcrafted features.
>
> >I think it helps to give more stats about each dataset presented in the benchmarks, e.g. number of samples etc.
>
> We have added the stats of each dataset in the Appendix (Table 9).

---

### Official Review · Reviewer_3WpQ · 2021-07-03
**Exciting push forward in the map from low-dimensional to 3D modeling of molecular structures**

**Rating:** 9
**Confidence:** 3
**Correctness:** The evaluation methods and design of …
**Clarity:** The paper is very well written.

**Strengths:**

-- This dataset will be useful to accelerate research in a very important area of biology/biophysics.
-- This reviewer understands that the data and code are readily and openly accessible.
-- Investigators have been trying to predict 3D structure for decades. A large fraction of these efforts have been based on “basic physics principles” or even protocols that engage large numbers of humans. More recently, many heavy players in machine learning including DeepMind have become interested in this problem. The availability of benchmarks can provide a major boost to the field.


**Weaknesses:**

-- I am not sure that this is a serious weakness but it seems that the field of 3D molecules is extremely large and heterogeneous. Predicting the structure of a 300 aminoacid protein is rather different from predicting the structure of a 20-atom ligand and the principles behind how chains of aminoacids folds can also be very different from those underlying folding of RNA molecules. The authors briefly allude to different subproblems (e.g. small molecule, protein, residue, etc.). I cannot help but wonder whether the field would benefit more from multiple separate datasets for different types of problems.

**Additional Feedback:**

None

**Documentation:**

The documentation is sufficient and clear.
It is the understanding of this reviewer based on the text that the data are available for general use.



**Ethics:**

No ethical concerns

**Relation To Prior Work:**

This is the main weakness of this reviewer (not the authors or the paper). I am not an expert in all the datasets for 3D molecular structures. The ones I know about are cited here, but I cannot claim completeness. Other reviewers may have more to say about this important point.

**Summary And Contributions:**

This paper introduces code and benchmarks that facilitate the use of state-of-the-art machine learning tools like convolutional neural networks and graph neural networks in the prediction of 3D molecular structures. Accurately modeling molecules in 3D is critical to mapping structure to function and has major implications to elucidate molecular mechanisms in biology and for drug discovery. The authors here provide a suite of readily accessible benchmark datasets and open-source code to help accelerate research and comparisons of computational models.

---

> ### Author Response · Authors · 2021-07-14
> **Reviewer Response**
>
> We thank the reviewer for their excitement and comments!  We respond to their comment regarding the heterogeneity of 3D molecular learning datasets below.
>
> > I am not sure that this is a serious weakness but it seems that the field of 3D molecules is extremely large and heterogeneous. Predicting the structure of a 300 aminoacid protein is rather different from predicting the structure of a 20-atom ligand and the principles behind how chains of aminoacids folds can also be very different from those underlying folding of RNA molecules. The authors briefly allude to different subproblems (e.g. small molecule, protein, residue, etc.). I cannot help but wonder whether the field would benefit more from multiple separate datasets for different types of problems.
>
> The reviewer is right that there is a huge variety in the scales of systems relevant in molecular science. One of the intentions in the design of ATOM3D actually is to span a large part of these scales. We think it will be interesting to see whether a type of method will emerge that is readily available across the entire spectrum of tasks. Even if not, it is not a priori clear where the limits of applicability for each new method are. We see ATOM3D as a framework in which these limits can be explored without much additional effort.

---

### Official Review · Reviewer_TL5E · 2021-07-04
**comprehensive work on dataset and benchmark for 3D molecular learning**

**Rating:** 9
**Confidence:** 3
**Clarity:** yes

**Strengths:**

- This paper is well motivated and clearly presented.
- Provides a landscape of important tasks/datasets, and state-of-the-art results. It will be a good start point for ML people interested to work on molecular learning.
- The dataset and models are well documented and open sourced. Also dataset creation and models are detailed in supp materials.
- The limitation of the current work and open questions are also discussed.

**Weaknesses:**

- The implementation of benchmark 3D models are indeed simple, but not sure they are "strong". It is not clear for a reader outside the community to know how strong the performance is if they are not familiar with the metrics. e.g. how far we are to solve a task. Also there are no 1d/2d models are in 2020?
- Compute / model sizes / inference time are not compared between models. I guess overall this domain are more data limited rather than compute limited.
- Lack of deeper insight regarding why 3D representation is better, e.g. what type of high-level structural pattern those 3D models can learn while 1D or 2D can't.

**Additional Feedback:**


- may provide a leaderboard for those tasks in the future
- Figure 1: PPI --> PIP
- L739: HDF5 --> LMDB
- L854 - 862: not clear to me after going through several times

**Correctness:**

- The claim that 3D representation is better than 1D/2D repr for tasks related to structure/shape is quite intuitive, as they provide more complete info. I don't doubt this. The challenge is probably we don't always have those 3D structures available.
- I like the way to split training/test data with 30% seq identity. The authors provide sufficient details on source, preprocessing and dataset code.
- For benchmark models, for some models e.g. Cormorant, reduced dataset are used, in this case how to compare with 1D/2D baseline, with the same training/test data?

**Documentation:**

well documented, open sourced, dataset easily downloadable. Looks like this is also a living dataset which welcomes new tasks/dataset/models.

**Ethics:**

Datasets are all curated from public database. Discussed dataset are biased towards solved, static structures. It's hard to interpret model predictions.

**Relation To Prior Work:**

not sure if related work section is complete. e.g. related work on protein folding.

**Summary And Contributions:**

This paper advocates 3D representation for solving a suite of prediction tasks related to small molecules and/or biopolymers. The author curated either existing or new dataset for each task with a focus on reducing similarity between train and test sequences. Simple yet robust implementation of 3 classes of 3D models are benchmarked against state-of-the-art models with 1D or 2D representation, and show better performance over lower-dim counterparts. The author provides observation and insights about how the choice of model class depends on the task at hand and the many-body hypothesis as a design guideline.
By providing benchmarking datasets/tasks and baseline models, this work will help lower the entry barrier for ML people to get into comp bio / molecule design.

---

> ### Author Response · Authors · 2021-07-14
> **Reviewer Response**
>
> We thank the reviewer for their insightful comments!  We answer the reviewer’s questions and comments below.
>
> > The implementation of benchmark 3D models are indeed simple, but not sure they are "strong". It is not clear for a reader outside the community to know how strong the performance is if they are not familiar with the metrics. e.g. how far we are to solve a task.
>
> Thank you for pointing out this source of confusion.  We intended to use the word “strong” to indicate these methods were often competitive with state-of-the-art, not that they had solved the tasks in an absolute sense.  We have rephrased the text to clarify and are more clearly linking to Table 7, where we delve into the overall state-of-the-art on each of these tasks, and often show competitive or superior performance.
>
> > Also there are no 1d/2d models are in 2020?
>
> For the PIP task, we find no models since Sanchez-Garcia et al. 2019 [1] that do not incorporate 3D information. For RES and MSP, there have been a few masked language models published since TAPE [2], e.g. ESM-1b [3], but we choose TAPE as a baseline because of its readily available implementations for training and their use of a more stringent split (heldout families for TAPE vs. 50% sequence identity for ESM-1b). For LBA and LEP, DeepCDA [4] have also recently proposed an approach using the 1D protein sequence and the 2D ligand graph and reported slightly better performance than DeepDTA and DeepAffinity, although they do not report results on PDBBind. It would indeed be interesting to run the comparison with this model on our datasets, although we don’t anticipate that our results will change dramatically.
>
> > Compute / model sizes / inference time are not compared between models. I guess overall this domain are more data limited rather than compute limited.
>
> This is correct.  We find that the main practical limitation for these datasets are the data, not the compute—all our models were trained on a single GPU.
>
> > Lack of deeper insight regarding why 3D representation is better, e.g. what type of high-level structural pattern those 3D models can learn while 1D or 2D can't.
>
> We agree that there is still much to learn about why these 3D representations are beneficial.  Through ATOM3D we hope to enable these future lines of work.
>
> > The claim that 3D representation is better than 1D/2D repr for tasks related to structure/shape is quite intuitive, as they provide more complete info. I don't doubt this. The challenge is probably we don't always have those 3D structures available.
>
> This is a good point.  With the rise of accurate computational modeling of 3D biomolecular structure (e.g., AlphaFold), we anticipate this will be less of an issue moving forward.
>
> > For benchmark models, for some models e.g. Cormorant, reduced dataset are used, in this case how to compare with 1D/2D baseline, with the same training/test data?
>
> In some cases of 3DCNN and ENN training, we only use a subset of the atoms in each structure. We describe in the appendix which reduced representations are used for each method. As we retain the most important atoms in each structure (within a radius around the region of interest, e.g., ligand or mutation site), the effect of the omitted atoms is likely minor. This kind of omission would be difficult to mimic in 1D/2D methods as it breaks sequences and bond graphs on which these methods are built.
>
> In two cases (ENN for RES and GNN for PIP), we use only a subset of structures from each dataset. These are subsampled randomly so they are representative for their respective split.
>
> The baselines always use the full dataset and the entire structure. Thus, when comparing to the baselines, the reported values for 3D methods likely represent a lower bound for their respective performance.
>
> > not sure if related work section is complete. e.g. related work on protein folding.
>
> Due to the broadness of the field, there is indeed a lot of prior work which we were not able to reference, including advances in protein structure prediction using deep learning. We therefore focused on discussing related work for the eight specific tasks we address in ATOM3D. For a review of recent deep learning methods in protein representation and folding, we refer to the work of Laine et al., 2021 [5].
>
> > may provide a leaderboard for those tasks in the future
>
> This is a good idea!  We will work to include this.
>
> > Figure 1: PPI --> PIP
> > L739: HDF5 --> LMDB
> > L854 - 862: not clear to me after going through several times
> We fixed the above and have rephrased this section to clarify our procedure.

---

> > ### Author Response · Authors · 2021-07-14
> > **References**
> >
> > References:
> >
> > 1. Sanchez-Garcia, R., Sorzano, C. O. S., Carazo, J. M., & Segura, J. (2019). BIPSPI: a method for the prediction of partner-specific protein–protein interfaces. Bioinformatics, 35(3), 470-477.
> > 2. Rao, R., Bhattacharya, N., Thomas, N., Duan, Y., Chen, X., Canny, J., ... & Song, Y. S. (2019). Evaluating protein transfer learning with TAPE. Advances in neural information processing systems, 32, 9689.
> > 3. Rives, A., Meier, J., Sercu, T., Goyal, S., Lin, Z., Liu, J., ... & Fergus, R. (2021). Biological structure and function emerge from scaling unsupervised learning to 250 million protein sequences. Proceedings of the National Academy of Sciences, 118(15).
> > 4. Abbasi, K., Razzaghi, P., Poso, A., Amanlou, M., Ghasemi, J. B., & Masoudi-Nejad, A. (2020). DeepCDA: deep cross-domain compound–protein affinity prediction through LSTM and convolutional neural networks. Bioinformatics, 36(17), 4633-4642.
> > 5. Laine, E., Eismann, S., Elofsson, A., & Grudinin, S. (2021). Protein sequence-to-structure learning: Is this the end (-to-end revolution)?. arXiv preprint arXiv:2105.07407.

---

### Decision · Program_Chairs · 2021-07-27

**Decision:**

Accept

**Comment:**

The authors propose datasets and benchmarks for developing machine learning novel algorithms on 3D representations of molecules. The reviewers note the importance of the problem, the clarity of the presentation, and the scope of the work. They strongly recommend acceptance.